

# Enhancing sentiment analysis of online comments: a novel approach integrating topic modeling and deep learning

Yan Zhu[1], Rui Zhou[1], Gang Chen[2] and Baili Zhang[3]

[1] School of Design and Art, Shanghai Dianji University, Shanghai, Shanghai, China
[2] Infrastructure Technology Business Group, Ant Group, Hangzhou, Zhejiang, China
[3] School of Computer Science and Engineering, Southeast University, Nanjing, Jiangsu, China

## ABSTRACT

Traditional statistical learning-based sentiment analysis methods often struggle to effectively handle text relevance and temporality. To overcome these limitations, this paper proposes a novel approach integrating Latent Dirichlet Allocation (LDA), Shuffle-enhanced Real-Valued Non-Volume Preserving (RealNVP), a double-layer bidirectional improved Long Short-Term Memory (DBiLSTM) network, and a multi-head self-attention mechanism for sentiment analysis. LDA is employed to extract latent topics within comment texts, revealing text relevance and providing fine-grained user feedback. Shuffle enhancement is applied to RealNVP to effectively model the distribution of text topic features, enhancing performance while avoiding excessive complexity in model structure and computational overhead. The double-layer bidirectional improved LSTM, through the coupling of forget and input gates, captures the dynamic temporal changes in sentiment with greater flexibility. The multi-head self-attention mechanism enhances the model's ability to select and focus on key information, thereby more accurately reflecting user experiences. Experimental results on both Chinese and English online comment datasets demonstrate that the proposed integrated model achieves improved topic coherence compared to traditional LDA models, effectively mitigating overfitting. Furthermore, the model outperforms single models and other baselines in sentiment classification tasks, as evidenced by superior accuracy and F1 scores. These results underscore the model's effectiveness for both Chinese and English sentiment analysis in the context of online comments.

## INTRODUCTION

In recent years, the exponential growth of User-Generated Content (UGC) (*Ali et al., 2022*; *Murshed et al., 2022*) on platforms like social media, forums, and e-commerce sites has opened new avenues for a deeper understanding of user experiences. This wealth of UGC, rich in emotional expressions, holds significant potential for informing decision-making processes across domains such as brand reputation management (*Machado, Miranda & Baldi, 2022*), market trend forecasting (*Li et al., 2022b*), and public policy development (*Fan*

Corresponding author
Rui Zhou,
23600006140103@st.sdju.edu.cn

*et al., 2021*). Nevertheless, the accurate and efficient extraction of emotional information from the massive volumes of UGC text data presents a persistent challenge.

Traditional sentiment analysis approaches, including rule-based methods (*Lin & Wu, 2022*) and those rooted in statistical learning (*Liu et al., 2023*), demonstrate some efficacy in processing straightforward text sentiment. However, they often struggle to account for the complexities of text relevance and temporality. Text relevance refers to the interconnectedness of texts in terms of topic, content, or context (*Li et al., 2023*), while temporality denotes the dynamic nature of sentiment over time (*Wang, 2021*). These two factors are particularly salient in UGC contexts such as social media and online comments, where users frequently engage in topic-focused discussions, and sentiment trajectories can shift as conversations unfold.

To address the challenges of text relevance and temporality in sentiment analysis, this study proposes a novel LDA-RealNVP-DBiLSTM-MHSA approach that integrates Latent Dirichlet Allocation (LDA) (*Zhang & Zhang, 2021*), Shuffle-enhanced Real-Valued Non-Volume Preserving (RealNVP) (*Draxler, Schnörr & Köthe, 2022*), a double-layer bidirectional improved Long Short-Term Memory (DBiLSTM) network (*Dellal-Hedjazi & Alimazighi, 2022*), and Multi-Head Self-Attention mechanism (MHSA) (*Raghavendra et al., 2022*). LDA, a well-established topic modeling technique, effectively reveals latent topic structures within text data, thus elucidating underlying relationships among texts. However, LDA assumes a static document collection, neglecting temporal changes and thus failing to capture topic evolution over time.

To overcome LDA's limitations and effectively bridge the gap between LDA-extracted topic features and BiLSTM's temporal modeling, we innovatively introduce Shuffle-enhanced RealNVP as a feature transformation and enhancement module. RealNVP, a flow-based generative model, learns reversible transformations to map high-dimensional, sparse topic features from LDA's output into a continuous, low-dimensional latent space, effectively addressing feature sparsity and reducing computational complexity for the subsequent BiLSTM model. Crucially, the Shuffle operation randomly shuffles feature dimensions, disrupting potential local correlations in the original feature space and enhancing the model's ability to model long-range dependencies in high-dimensional, complex data, thereby improving feature representation robustness (*Dinh, Sohl-Dickstein & Bengio, 2016*). Furthermore, RealNVP's flow-based nature enables feature generation, enriching the input to BiLSTM by directly concatenating generated features with the original topic features. The implicit regularization effect of Shuffle also helps mitigate overfitting and enhances training stability. Additionally, fixing random seeds ensures experiment reproducibility, while hyperparameter optimization further controls the impact of randomness.

To enhance the handling of text temporality, we introduce a structurally optimized double-layer BiLSTM model. By coupling the forget gate and input gate, this architecture achieves enhanced information processing efficiency. The deep bidirectional processing mechanism of the double-layer BiLSTM structure enables it to capture complex patterns and long-term dependencies in sequential data better, modeling temporal information in the text at a deeper level, particularly the complex changes in sentiment over time. The

choice of a double-layer configuration strikes a balance between model complexity and operational efficiency, as excessive BiLSTM layers can introduce gradient vanishing or explosion problems, increasing training difficulty without necessarily yielding significant performance gains. Moreover, the introduction of the Multi-Head Self-Attention mechanism allows the model to capture dependencies between any two positions in the sequence, better modeling long-range dependencies and contextual information in the text, which is crucial for semantic understanding and sentiment capture in sentiment analysis tasks (*Li et al., 2020*).

The primary research questions addressed in this study are as follows:

(1) How to construct a sentiment analysis model that integrates LDA, Shuffle-enhanced RealNVP, double-layer BiLSTM, and MHSA to effectively address the challenges of relevance and temporality in UGC text data;

(2) How to effectively transform the output of the LDA model into the input of the BiLSTM model, achieving seamless integration between the models through Shuffle-enhanced RealNVP;

(3) How to construct and optimize a double-layer BiLSTM structure to better capture temporal information in the text.

The remainder of this paper is structured as follows: 'Related work' reviews existing research on text sentiment analysis, emphasizing applications and achievements of LDA, RealNVP, and LSTM models in this field. 'Methodology' details the methodology, covering the integration strategy of LDA, enhanced RealNVP, double-layer BiLSTM, and MHSA, model selection, and other core aspects to ensure experimental rigor. 'Experiments' presents the LDA-RealNVP-DBiLSTM-MHSA model construction, detailing dataset selection, parameter settings, and evaluation metrics. 'Results and Discussion' conducts a comparative analysis of datasets, validating the proposed model's superiority over traditional methods and benchmarks. 'Conclusions' concludes by summarizing the study's contributions and outlining future research directions.

## RELATED WORK

### Text sentiment analysis

Text sentiment analysis, a pivotal subfield of natural language processing (NLP), aims to automatically identify and classify the subjective polarity (positive, negative, or neutral) expressed in textual data through computational techniques (*Deng et al., 2021*). This technology has become increasingly vital for understanding public opinion and tracking the evolution of societal sentiment. The field has witnessed a methodological evolution, progressing from traditional lexicon-based approaches to machine learning and, more recently, deep learning techniques (*Chen et al., 2023b*).

In text sentiment analysis, the LDA model has been widely recognized for its robust feature extraction capabilities. It can identify latent topics within textual data, facilitating the optimization of information resources (*Wang & Wu, 2023*). Simultaneously, the LSTM model, a prominent deep learning technique, excels at processing sequential data. It can directly and accurately predict sentiment labels in text, with applications spanning stock

index prediction (*Xu & Tian, 2021*) and enhancing intelligent library services (*Zhao et al., 2023*). Both LDA and LSTM models offer unique strengths and when combined, have the potential to advance the field of text sentiment analysis.

Enhancing the comprehensiveness and accuracy of text sentiment analysis is an active area of research, with a particular focus on the fusion and complementarity of multiple models (*Chen & Guo, 2024*; *Zhu et al., 2022*). The emergence of numerous datasets and the application of multimodal methods have provided ample impetus for this trend (*Zhu et al., 2023*). For example, *He et al. (2023)* successfully integrated Bidirectional Encoder Representations from Transformers (BERT) with the LDA model to improve sentiment classification. *Deng et al. (2021)* combined convolutional neural networks, bidirectional LSTMs, and attention mechanisms for enhanced sentiment analysis. *Liu et al. (2022)* incorporated LDA into convulational neural network (CNN) to improve short-text fault information classification. Additionally, models such as Robustly Optimized BERT Approach Bidirectional Long-short Term Memory Conditional Random Field (RoBERTa BiLSTM CRF) (*Xu et al., 2023*) have gained significant attention for their ability to effectively capture sentiment information in text, thereby improving classification accuracy and contributing to new directions in fine-grained sentiment analysis research.

## LDA theme model

The LDA model is a foundational technique in the field of text mining. By establishing a three-layer Bayesian probabilistic framework linking vocabulary, topics, and documents, LDA uncovers the latent thematic structure inherent within text data. The model operates under the assumption that the vocabulary within a document is drawn randomly from latent topics, allowing it to capture the underlying complexity of the text while ensuring topic coherence (*Xue et al., 2024*). As depicted in Fig. 1, the LDA process involves several key steps. Initially, a topic distribution $\theta_i$ is generated for each document through sampling from a Dirichlet distribution $\alpha$, thus laying the groundwork for the document's thematic content. Subsequently, each word's topic affiliation is determined by sampling from a multinomial distribution over topics. Next, a word distribution $\varphi_{Zi,j}$ is generated for each topic through another Dirichlet distribution $\beta$. Finally, the actual words comprising the document are generated by sampling from a multinomial distribution over words, effectively revealing the underlying topic structure of the text (*Hu, Han & Wang, 2024*).

The LDA model has found widespread application in the realm of text sentiment analysis. *He, Zhou & Zhao (2022)* employed LDA to conduct an in-depth analysis of user comments, extracting sentiment-laden topics that influence user experiences and offering recommendations for product and service enhancements. In the context of social media sentiment analysis, *Wang, Sun & Wang (2022)* leveraged LDA to precisely identify topics discussed by the public and uncover their corresponding sentiment orientations. Moreover, LDA has been utilized to analyze user comments from diverse settings such as museums and theme parks, effectively identifying areas of concern for users and providing actionable insights for design improvements to management (*Fudholi et al., 2023*).

As the complexity of text sentiment analysis increases, the limitations of using LDA models in isolation have become apparent. Consequently, the development of LDA fusion

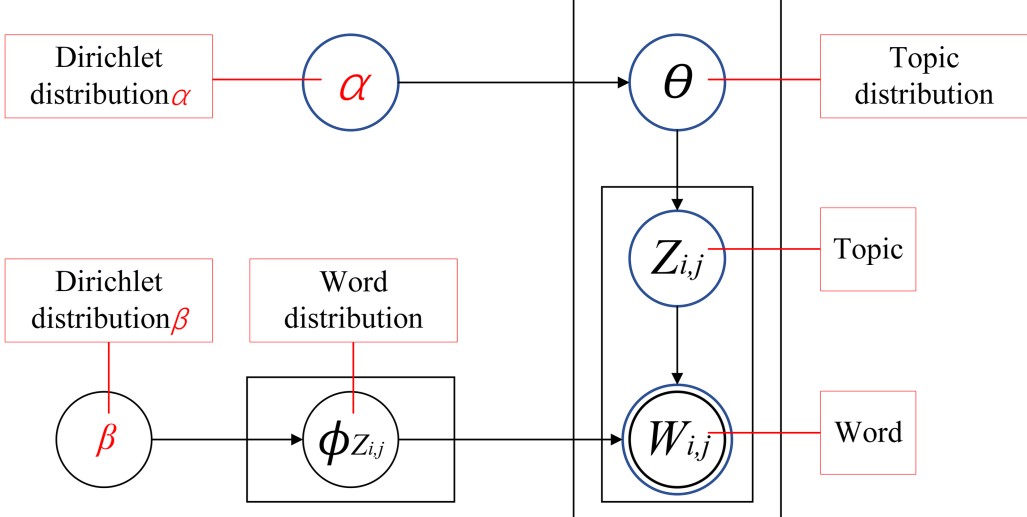

**Figure 1** **LDA topic model structure diagram.**

models, which combine LDA with other machine learning or deep learning models, has emerged as a crucial strategy for enhancing performance. This approach has demonstrated significant improvements in topic classification accuracy and sentiment analysis. For instance, *Watanabe & Baturo (2024)* proposed a model integrating seed words with sequential LDA modeling, resulting in increased precision in topic matching and enhanced stability of sentence-level topic transitions. *Wu & Shen (2024)* improved the accuracy and foresight of LDA in detecting emerging topics by optimizing evaluation methods and constructing a novel identification framework. Additionally, *Wu, Du & Lin (2023)* integrated the Neural Basis Expansion Analysis (N-BEATS) network model with LDA, improving the accuracy of technological theme prediction. These fusion models highlight the ongoing relevance and innovative potential of LDA-based approaches in text sentiment analysis.

### Normalized flow's RealNVP model

Normalized Flows (NFs) (*Dias et al., 2020*), exemplified by the RealNVP model, have emerged as a powerful class of generative models capable of transforming simple distributions into complex data distributions through a series of invertible transformations. The core strength of RealNVP lies in its coupling layers, which partition and affine-transform input data, enabling efficient information mixing and the generation of intricate distributions (*Papamakarios et al., 2021*). This capacity for complex distribution modeling has inspired the exploration of leveraging a shuffled RealNVP variant to enhance feature representations in the context of text analysis, distinct from the broader application of NFs.

The potential of NFs in capturing nuanced emotional expressions within textual data has been demonstrated in sentiment analysis. *Tran et al. (2019)* pioneered the application of NFs to discrete data through their discrete flow model, facilitating effective text sentiment analysis and sequence dependency modeling. Building on this, *Li, Wu & Wang (2020)*

introduced the Flow-Emotion model, leveraging RealNVP to learn sentiment embedding spaces and incorporating attention mechanisms for context-aware modeling, further underscoring the efficacy of NFs in sentiment analysis. While these works highlight the power of NFs in capturing textual nuances, they primarily focus on sentiment classification and do not explore the integration of NFs with topic modeling techniques like LDA, a gap addressed in this research.

Beyond sentiment analysis, RealNVP has found extensive application in modeling temporal data. *Papamakarios, Pavlakou & Murray (2017)* combined autoregressive structures and masking with RealNVP in their Masked Autoregressive Flow (MAF) model, establishing a foundation for efficient sequential modeling and the application of NFs in time series analysis. *Kim et al. (2022)* extended RealNVP to the financial domain, enabling fast pricing of path-dependent exotic options by simulating stochastic volatility models. Further highlighting RealNVP's aptitude for handling complex time series, *Kobyzev, Prince & Brubaker (2021)* and *Zhao et al. (2024)* achieved significant results in anomaly detection with their Conditional Flow for Anomaly Detection (CFlow-AD) and Deep Denoising Autoencoder Normalizing Flow (DDANF) models, respectively, by integrating RealNVP with conditional normalized flows and denoising autoencoders. These applications showcase the versatility of RealNVP in diverse domains; however, its potential for enhancing feature representations derived from topic models like LDA for downstream tasks like LSTM-based text analysis remains largely unexplored, a direction pursued in this study.

## LSTM model

The LSTM model, an advanced recurrent neural network architecture, excels in processing sequential data due to its unique design. As depicted in Fig. 2, the LSTM model incorporates memory cells and three logical gates: the input gate, forget gate, and output gate. This architecture enables the LSTM to effectively capture and process long-term dependencies, making it particularly suitable for tasks requiring contextual understanding (*Yu et al., 2023*). By seamlessly connecting long-term memory with current input, LSTM addresses the challenges of long-term dependencies and gradient vanishing that often plague traditional recurrent neural networks.

The LSTM model's exceptional performance in text sentiment analysis is attributed to its ability to capture sentence structure and semantics, making it a valuable tool for sentiment classification tasks. By integrating deep learning with time series analysis, LSTM leverages contextual information effectively, thereby enhancing the model's scalability and performance (*Dellal-Hedjazi & Alimazighi, 2022*; *Liu, Wang & Li, 2023*). For instance, a text sentiment polarity classification method based on an improved residual neural network (ResNet) and LSTM has demonstrated increased classification accuracy while maintaining operational efficiency (*Liu, Yang & Yu, 2023*). Furthermore, *Xuan & Deng (2023)* enhanced LSTM's ability to capture textual sentiment by incorporating a deep attention mechanism and topic embedding strategy.

In scenarios involving complex, long texts, a single LSTM model may fail to retain crucial information during processing. Consequently, multi-model fusion has emerged as

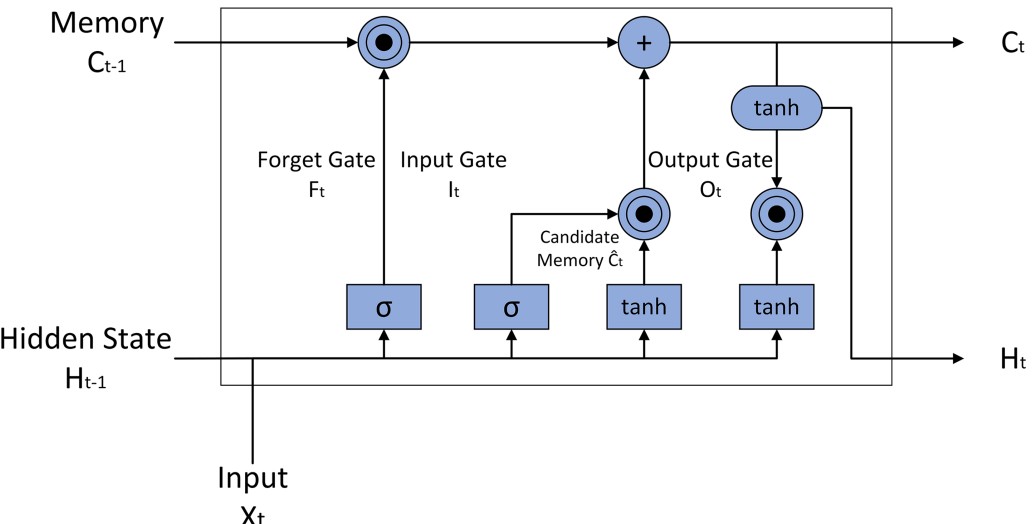

**Figure 2  LSTM model structure diagram.**

a key strategy for enhancing performance. *Khan et al. (2022a)* proposed a deep learning model integrating CNN with LSTM, achieving notable success in sentiment analysis of social media texts. This hybrid model first extracts local features from the text using CNN and subsequently processes temporal dependencies using LSTM, effectively combining the strengths of both architectures: CNN for feature extraction and LSTMs for temporal modeling.

Numerous studies have explored the fusion of LDA and LSTM models. For instance, *Zeng, Yang & Zhou (2022)* employed a combination of LDA and the Attention-BiLSTM model to reveal the dynamics of online public opinion. *Li et al. (2022a)* integrated LDA, BiLSTM, and the Self-attention model, achieving enhanced accuracy in short text classification. *Hu et al. (2022)* proposed a CNN-BiLSTM-MHSA-based electroencephalography (EEG) emotion recognition method, utilizing CNN to extract local features, BiLSTM to establish a temporal emotion change model, and MHSA to enhance the focus on key features, significantly improving emotion recognition rates on the DEAP dataset. These studies underscore the potential of combining models for improved text classification and sentiment analysis.

## METHODOLOGY

Failure to consider the relevance and temporal context of texts during analysis can result in misinterpretations of sentiment within specific contexts and its trajectory over time. In this study, we propose a refined approach that integrates topic modeling, feature enhancement, deep sequential modeling, and attention mechanisms to achieve a more comprehensive and accurate capture of sentiment in UGC texts. This enhanced model demonstrates better performance in preserving sparse features inherent in short texts and extracting salient features from information-rich texts.

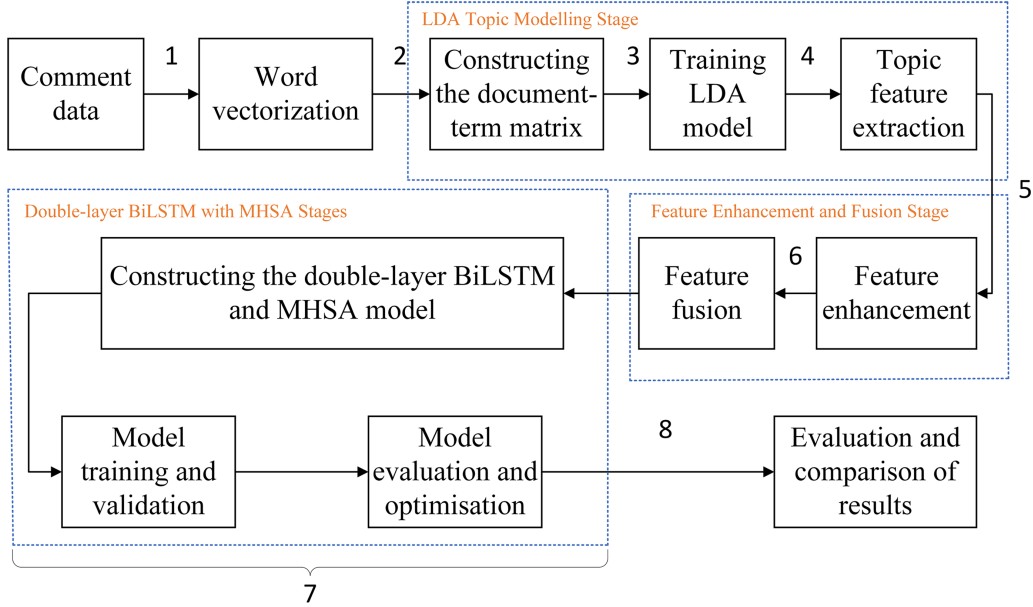

**Figure 3   Experiment overall flow chart.**

The proposed LDA-RealNVP-DBiLSTM-MHSA model follows the process illustrated in Fig. 3.

(1)   Comment texts are converted into word vectors for machine processing;

(2)   The word vectors are input into the LDA model to create a document-term matrix;

(3)   The LDA model is trained to obtain the topic distribution for each document;

(4)   Topic features are extracted from the trained LDA model, and a dataset is constructed that reflects the topical characteristics of each document;

(5)   A Shuffle-enhanced RealNVP model is built to obtain an enhanced topic feature dataset;

(6)   The enhanced topic feature dataset and the original word vector sequence are directly concatenated to obtain a fused feature representation dataset;

(7)   A double-layer BiLSTM and MHSA model is built, and the effectiveness of the proposed method is validated;

(8)   Model results are evaluated and compared.

Figure 4 illustrates the detailed internal structure of the proposed model, depicting the complete process from data preparation and word vectorization, through LDA model processing and RealNVP feature enhancement, to the double-layer BiLSTM and MHSA model processing. The Shuffle operation, by randomly shuffling the order of feature dimensions, breaks the potential local correlations in the original feature space, enhancing the model's ability to model long-range dependencies in high-dimensional complex data, thereby improving the robustness of feature representation. Moreover, the flow model characteristic of RealNVP endows it with generative capabilities. By directly concatenating the generated features with the original topic features, the feature representation input to BiLSTM is further enriched. Within the double-layer improved BiLSTM structure, the

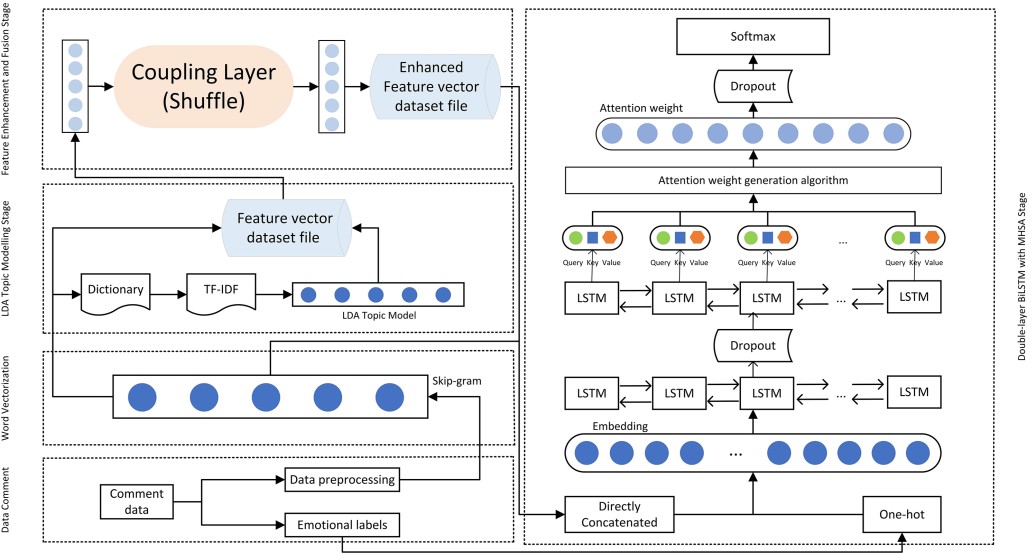

**Figure 4** The internal structure of the model.

first BiLSTM layer is responsible for initial temporal feature extraction while preserving sequential information integrity. The second BiLSTM layer then performs deeper feature extraction, compressing the entire sequence into a fixed-size feature vector suitable for subsequent classification or other downstream tasks. This stacked BiLSTM configuration enables the model to learn intricate sequence features and contextual relationships. By coupling the forget gate and the input gate, the processing time of the double-layer BiLSTM is effectively reduced, and the accuracy of processing is ensured. MHSA empowers the model to attend to multiple representation subspaces simultaneously, thereby enhancing its understanding and modeling of complex semantics. Furthermore, dropout regularization is applied to mitigate overfitting and enhance the model's generalization capability.

## Word vectorization

Word vectorization, a fundamental technique in NLP, converted words into fixed-dimensional vectors, encapsulating semantic relationships and reducing feature dimensionality (*Hu, Han & Wang, 2024*). This study employed the Skip-gram method (*Xia et al., 2020*) for word vectorization, the principle of which is illustrated in Fig. 5. The core of Skip-gram lies in predicting a word's context words to capture their inherent connections. Utilizing the preprocessed document set $S_t$ as input, word vectors were generated using Skip-gram. The similarity between word vectors was quantified by calculating their inner product, thus inferring the correlation between words. Specifically, the similarity between any two-word vectors, $W_i$ and $W_j$, was given by:

$$\text{Similarity}\left(W_i, W_j\right) = W_i \cdot W_j = \sum_{k=1}^{n} W_{ik} \cdot W_{jk}. \tag{1}$$

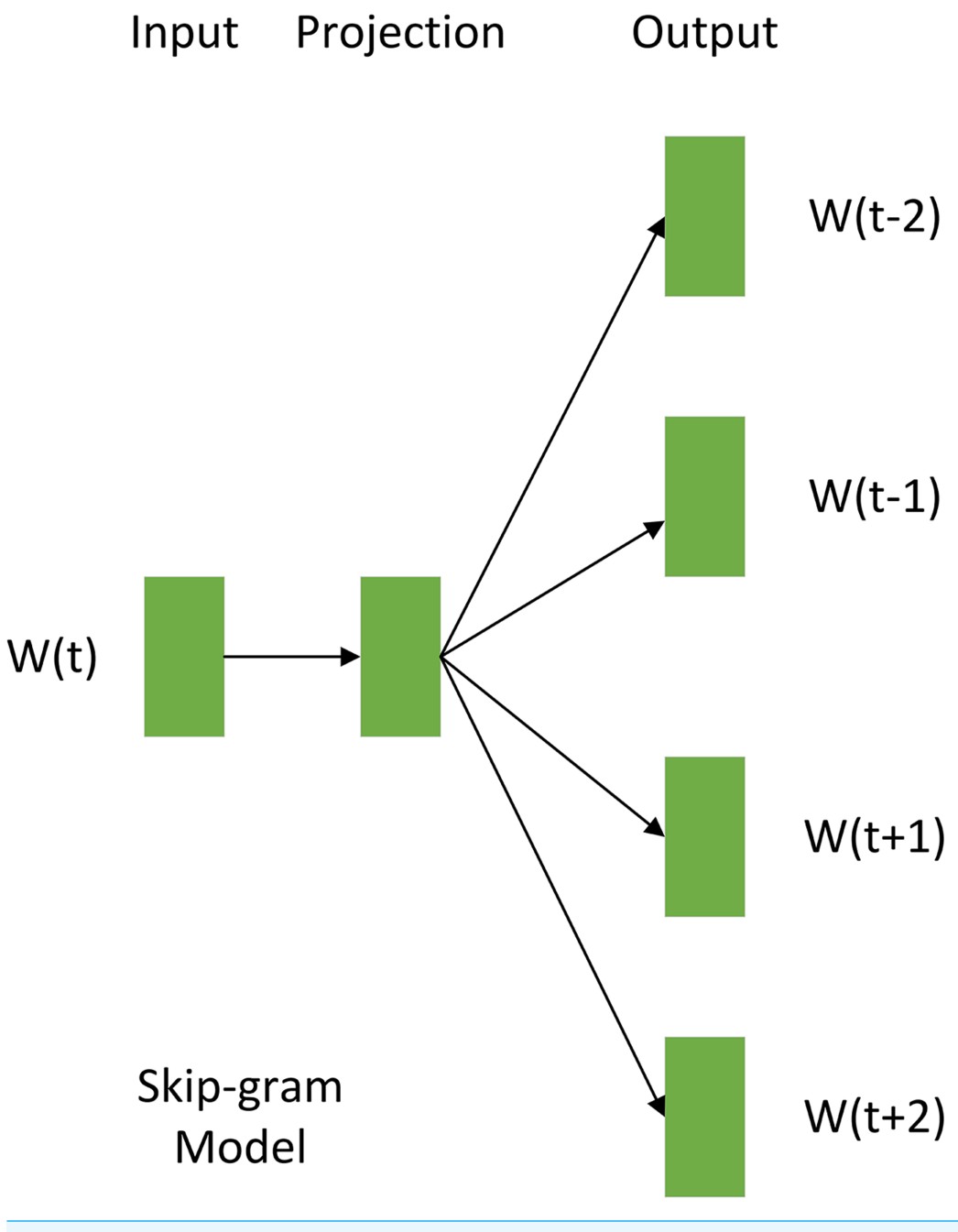

**Figure 5  Skip-gram method schematic.**

Where $W_i$ and $W_j$ represented the vector forms of the two words, $n$ was the dimensionality of the word vector, and $W_{ik}$ and $W_{jk}$ represented the $k$-th component of $W_i$ and $W_j$, respectively.

Furthermore, this study employed the softmax function to compute the probability distribution of each word in the vocabulary being predicted given a central word. The

softmax function is defined as follows:

$$P\left(W_j|W_i\right) = \frac{exp\left(W_i \cdot W_j\right)}{\sum_{k=1}^{m} exp\left(W_i \cdot W_k\right)}. \tag{2}$$

Where $P(W_j|W_i)$ represented the probability of word $W_j$ being predicted given the central word $W_i$, and $m$ represented the total number of distinct words in the vocabulary.

The probability distribution of each word in the vocabulary being predicted, given a central word, was obtained by calculating the inner product between their corresponding word vectors and normalizing the results using the softmax function. Through this word to vector (Word2Vec) model processing, text data underwent vectorized transformation, converting high-dimensional textual information into a lower-dimensional set of word vectors, $D_t$. The Skip-gram word embedding method was chosen for its ability to capture nuanced semantic relationships between words, thereby improving the model's accuracy (*Shobana & Murali, 2021*).

As Fig. 5 illustrates, the Skip-gram model acquires distributed representations of words by predicting context words from a center word. This process maps words into a low-dimensional dense vector space, where each word vector encodes semantic information. Semantically similar words are positioned close to each other in this vector space, effectively capturing their relationships. Upon completion of Skip-gram training, each word is associated with a low-dimensional dense vector, and the collection of these vectors constitutes the word vector set $D_t$ (illustrated in Fig. 4). This dimensionality reduction of the text data furnishes structured semantic input for downstream models.

## LDA topic modelling stages
### Constructing the document-term matrix

Subsequently, the Dictionary module of the Gensim library (*Khan et al., 2022b*) was employed to convert the word-vectorized text dataset $D_t$ into a dictionary. This process involved traversing all tokenized documents, counting the frequency of each unique word, and establishing a mapping between words and their corresponding frequencies. Additionally, detailed word frequency statistics were recorded.

$$count(W) \leftarrow count(W) + 1. \tag{3}$$

To further refine the weight matrix, this study utilized the Term Frequency-Inverse Document Frequency (TF-IDF) algorithm (*Liu, Chen & Liu, 2022*). TF-IDF quantified the significance of each word within a specific document by multiplying its term frequency (TF) with its inverse document frequency (IDF). Term frequency denoted the frequency of a given word in the current document. Inverse document frequency, calculated as the logarithm of the ratio between the total number of documents and the number of documents containing the word (with one added to avoid division by zero), measured the word's rarity across the entire corpus.

$$TF - IDF = TF \times IDF. \tag{4}$$

$$TF = \frac{N}{M}. \tag{5}$$

$$IDF = log\left(\frac{X}{Y+1}\right). \tag{6}$$

The resulting document-term matrix, $R_t$, effectively addressed the challenge of sparse representation often encountered in textual data. By incorporating both term frequency within individual documents and inverse document frequency across the entire corpus, this approach offered a more comprehensive representation of word importance and document topic characteristics.

LDA models traditionally operate on a document-term matrix, with matrix elements quantifying term weight or frequency within each document. As depicted in Fig. 4, the LDA topic modeling phase begins by employing a dictionary to assign unique IDs to the terms in the word-vectorized $D_t$. Subsequently, it processes the TF-IDF matrix, representing each document as a bag-of-words consisting of term IDs and their corresponding TF-IDF values. Ultimately, the bag-of-words representations for all documents are consolidated into a document-term matrix $R_t$, serving as the input corpus for the LDA model, initiating the training process.

### Training the LDA model

To investigate the underlying structure of the text data, this study employed the LDA model to identify latent topics within the documents. To optimize model performance, data were read from the document-term matrix file $R_t$ and the LDA model was trained with adjusted parameters.

To identify optimal model parameters, this study employed perplexity and coherence as evaluation metrics (*Ran & Li, 2023*). Perplexity, a measure of how well a probability model predicts a sample, is a key indicator of model fit. It was calculated as follows:

$$Perplexity(D) = exp\left\{-\frac{\sum_{d=1}^{M} log\, p(w_d)}{\sum_{d=1}^{M} N_d}\right\}. \tag{7}$$

Where $D$ represented the test set in the corpus, containing $M$ reviews. $N_d$ denoted the number of words in review $d$, $w_d$ represented the words in review $d$, and $p(w)_d$ was the probability of generating the word $w_d$ according to the model. The lowest point, or a point close to the lowest point, on the perplexity curve was typically considered the optimal model configuration, as it indicated improved generalization ability.

In addition to perplexity, this study utilized a coherence curve to assess the semantic coherence among words within the model. The peak of the coherence curve typically signified the optimal number of topics, where word relationships were strongest and most meaningful.

By comprehensively considering the analysis results of perplexity and coherence curves, the optimal number of topics for the LDA model was determined. After establishing

the optimal configuration, the trained LDA model file $L_t$ and its evaluation results were produced. Utilizing perplexity and coherence curves not only enhanced the accuracy of LDA model selection but also ensured the model's effectiveness in capturing text associations and temporal dynamics. Moreover, these metrics served as crucial indicators for assessing the performance of subsequent models in the analysis pipeline.

### Topic feature extraction

In this study, the trained LDA model $L_t$ and the processed word vector dataset $D_t$ served as input for topic feature extraction. To ensure compatibility with the LDA model's input requirements, appropriate transformations were applied to the original data. Subsequently, the Gibbs sampling (*Zhao et al., 2021*) algorithm was employed for iterative approximation of the model's posterior distribution. In each iteration, new parameter values were sampled based on the current estimates. Core statistics, such as the mean and standard deviation of model parameters, were then computed from the numerous samples obtained.

In the topic feature extraction process, the number of topics, $k$, was initialized based on the analysis of coherence and perplexity curves. To determine the topic distribution for each document, the probability of observing the remaining words in a document, given the removal of a specific word, was calculated as follows:

$$P(Zi = k|w, z - i, \alpha, \beta) \propto \frac{nk, -i + \alpha}{n., -i + K\alpha} \times \frac{nw, k, -i + \beta}{nk, -i + V\beta}. \tag{8}$$

Where $nk$, $-i$ represented the number of words assigned to topic $k$ in the current document after excluding the $i$-th word; $n.$, $-i$ denoted the total number of topics assigned to all words in the current document after removing the $i$-th word; $nw$, $k$, $-i$ indicated the number of times word $w$ was assigned to topic $k$ after excluding the $i$-th word, and $V$ was the total number of unique words in the corpus. Through repeated iterations of Gibbs sampling until vocabulary convergence, the final output was a dataset file $E_t$ containing feature vectors of the topic distribution for each document.

The iterative Gibbs sampling and associated probability calculations facilitated the accurate extraction of topic features, revealing the underlying semantic structure of the text.

## Feature enhancement and fusion
### Feature enhancement with shuffle

This study introduces the RealNVP model to model and enhance the topic features generated by LDA. RealNVP's concise and reversible structure contributes to improved efficiency in modeling text topic features for sentiment analysis (*Dinh, Sohl-Dickstein & Bengio, 2016*). As illustrated in Fig. 6, the RealNVP model incorporates the Shuffle operation within its coupling layer to facilitate data mixing and bolster the model's generalization capabilities. The Shuffle operation randomly rearranges the concatenated vectors $h_1$ and $h_2$ from the flow output, where $s$ and $t$ are vector functions parameterized by neural networks (*Su, 2018*).

The reversibility inherent to the RealNVP model ensures lossless information transmission during transformation, providing high-quality feature representations to the

**Peer**J Computer Science

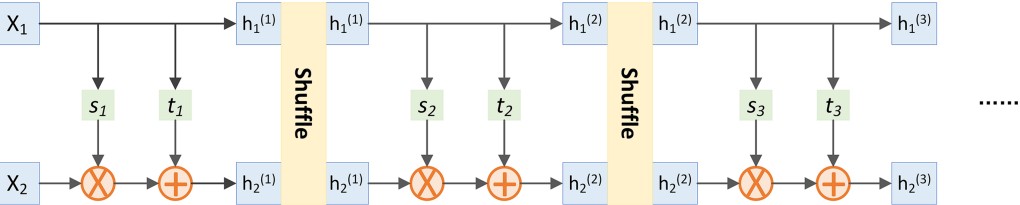

**Figure 6** RealNVP with Shuffle.

BiLSTM model. While preserving model reversibility, the Shuffle operation introduces data perturbation, thereby augmenting the model's adaptability and robustness for sentiment classification tasks. Experimental design and hyperparameter tuning mitigate potential instability arising from this randomness, ensuring stable model output. Furthermore, aligning the RealNVP model's hidden layer dimension with that of the LDA topic features guarantees effective data processing.

When applied to LDA's topic features, the RealNVP model enhances both expressive power and robustness, offering an effective feature enhancement strategy for sentiment analysis. The model's transformation is defined as:

$$y_{1:d} = x_{1:d}. \tag{9}$$

$$y_{d+1:D} = x_{d+1:D} \cdot exp(s(x_{1:d})) + t(x_{1:d}). \tag{10}$$

Where $s$ and $t$, parameterized by neural networks, implement nonlinear data transformations. The corresponding inverse transformation is:

$$x_{1:d} = y_{1:d}. \tag{11}$$

$$x_{d+1:D} = \left(y_{d+1:D} - t(x_{1:d})\right) \cdot exp(-s(x_{1:d})). \tag{12}$$

This transformation preserves data volume while achieving effective data mixing, furnishing the BiLSTM model with rich and uniform feature representations. By processing the LDA-processed text dataset $E_t$ with the RealNVP model's Shuffle coupling layer transformation, the data is encouraged to fully integrate contextual information, yielding an enhanced topic feature dataset $F_t$.

### Feature fusion

Direct concatenation was selected as the feature fusion strategy due to its simplicity, ability to maintain information integrity, and proven success in multi-modal tasks (*Xia, Li & Liu, 2023*). By directly concatenating the original word vector sequence $D_t$ with the enhanced topic feature vector $F_t$, both the fine-grained semantics of the text and the high-level topic information are preserved, avoiding the complexity and potential information loss associated with other fusion methods.

While LDA topic models excel at extracting the topic distribution of a text, they may overlook crucial semantic details such as word order and subtle nuances in meaning (*Liu &*

*Wei, 2020*). Word vector sequences, through their dense vector representations, effectively encode this information. Therefore, incorporating the original word vector sequence in the feature fusion stage compensates for the limitations of the topic model, providing the BiLSTM model with a richer and more comprehensive text representation, ultimately leading to improved sentiment classification accuracy.

By concatenating the original word vector sequence $D_t$ and the enhanced topic feature $F_t$, effective fusion of multi-modal features is achieved. This strategy not only enhances the model's expressive power, enabling it to capture a broader range of textual information, but also strengthens its understanding of textual sentiment, culminating in a dataset $N_t$ well-suited for BiLSTM input.

## LSTM sentiment analysis model
### Data preparation
In this study, a text dataset file $N_t$, resulting from feature fusion, and its corresponding sentiment labels were loaded. To ensure the model's effectiveness and generalization ability, a dynamic strategy was employed to partition the dataset into a training set $X_t$ and a testing set $C_t$.

To transform the sentiment labels into a numerical format suitable for model interpretation, one-hot encoding technology (*Gu & Sung, 2021*) was utilized. This technique represents each category as a vector, where only one element is 1 and the remaining elements are 0. The specific encoding scheme was as follows:

Positive evaluation $\rightarrow$ [1,0,0]
Neutral evaluation $\rightarrow$ [0,1,0]
Negative evaluation $\rightarrow$ [0,0,1]

By employing one-hot encoding, positive, neutral, and negative evaluations were converted into a numerical format that the model could process.

### Constructing a double-layer improved LSTM and MHSA model
To delve into the temporal and contextual information within the text data, a double-layer improved BiLSTM model was constructed, incorporating a MHSA to enhance the model's ability to capture complex emotional dynamics and semantic relationships. As shown in Fig. 4, the network architecture comprised an Embedding layer, two stacked improved BiLSTM layers (with a Dropout layer following each to mitigate overfitting), an MHSA layer, and a fully connected Dense layer. This design progressively extracted and processed text features, enabling the model to effectively capture the intricate nature and temporal dynamics of textual data.

In contrast to traditional sentiment analysis approaches that employ standard LSTM networks, this study utilized a modified LSTM variant featuring a coupled forget and input gate mechanism. The internal architecture of this modification is illustrated in Fig. 7.

In the standard LSTM architecture (Fig. 2), the forget gate's calculation is governed by the following equation:

$$F_t = \sigma\left(W_f \cdot \left[H_{t-1}, X_t\right] + b_f\right). \tag{13}$$

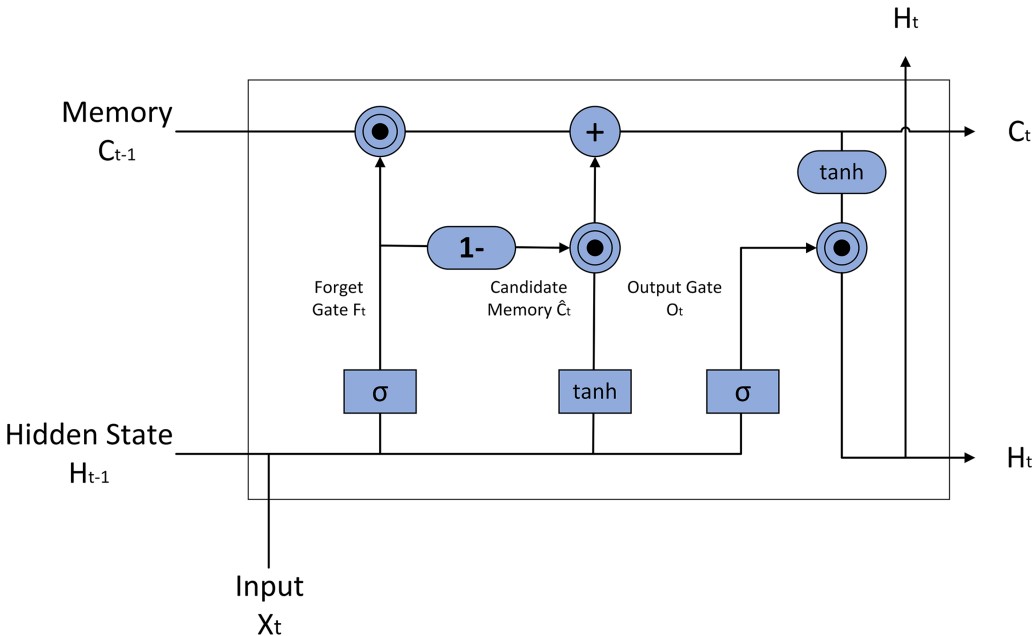

**Figure 7**   **Diagram of LSTM structure with coupled oblivion gate and input gate.**

In this equation, the sigmoid activation function ($\sigma$) compresses the input value into the range [0, 1], dictating the proportion of information from the previous cell state to be discarded. $W_f$, the forget gate's weight matrix, maps the input to the forget ratio, while $b_f$, the forget gate's bias vector, enables it to maintain a baseline activation level independent of input. The input gate's calculation proceeds as follows:

$$I_t = \sigma\left(W_i \cdot \left[H_{t-1}, X_t\right] + b_i\right). \tag{14}$$

$$\hat{C}_t = tanh\left(W_c \cdot \left[H_{t-1}, X_t\right] + b_c\right). \tag{15}$$

In this expression, $\hat{C}_t$ denotes the candidate cell state, representing the new information potentially added to the cell state in the current time step. However, the conventional LSTM architecture suffers from several drawbacks. First, due to the independent calculation of the forget and input gates, each gating mechanism requires separate weight matrices and bias vectors. This increases the model's parameter count, raising the risk of overfitting, particularly with limited training data. Second, each gating mechanism necessitates independent matrix operations and activation function calculations, increasing the model's computational complexity. This can become a performance bottleneck for large-scale data processing or real-time applications. Third, the separate computations of the forget and input gates may lead to redundant or conflicting decision outcomes. For example, the information retained by the forget gate might be similar to the new information added by the input gate, resulting in information redundancy. To mitigate these shortcomings, this study employed a coupled forget and input gate design, as depicted in Fig. 7. The

calculation is defined as follows:

$$C_t = F_t * C_{t-1} + (1 - F_t) * \hat{C}_t. \tag{16}$$

In this design, $1 - F_t$ can be interpreted as an implicit representation of the input gate. This coupling strategy allows the forget and input gates to not only share parameters but also operate synergistically in a complementary fashion. Consequently, this design not only streamlines the network's structural complexity but also improves the model's training efficiency and response speed by reducing redundant computations and the number of parameters, while maintaining accuracy.

The double-layer improved BiLSTM architecture enhanced the model's capacity to process complex text data, particularly long sequences with deep-level dependencies. The first BiLSTM layer extracted initial temporal features while preserving the output at each time step to ensure comprehensive temporal information transfer. The second BiLSTM layer further refined these features, focusing on capturing long-term dependencies within the text. During model construction, careful consideration was given to the dimensionality of each layer. The output dimension of the first BiLSTM layer was set to be sufficiently large to capture the rich features present in the text, while the second layer's output dimension was tailored for core feature extraction. The model's input dimension was aligned with the output dimension of the preceding LDA model to facilitate effective information flow.

Subsequently, MHSA was employed to comprehensively capture the intricate interactions within the text features. MHSA's multiple attention heads enable parallel processing of different parts of the input sequence, effectively capturing both global and local contextual information (*Xiao et al., 2020*). This multi-level attention mechanism not only enhances the model's sensitivity to global information but also facilitates a deeper understanding of semantic associations and contextual dependencies within the text, crucial for tasks like sentiment analysis. Compared to other self-attention mechanisms, MHSA's multi-head structure allows it to capture multi-scale features in parallel, improving the model's expressive power while mitigating the risk of overfitting through attention dispersion (*Xiao et al., 2020*). The multi-head mechanism empowers MHSA to simultaneously focus on various aspects and granularities of the input sequence, striking a balance between global semantics and local details. This multi-granularity feature extraction capability is essential for in-depth understanding of textual semantics, particularly for sentiment analysis. Additionally, the fusion of multi-head outputs further enhances the model's expressive power, enabling it to learn more complex and abstract text feature representations. As depicted in Fig. 4, MHSA takes the output sequence of the second BiLSTM layer as input and generates Query, Key, and Value vectors through linear transformations (*Tan et al., 2023*), which are then used to calculate attention weights.

$$Q = XW^Q, K = XW^K, V = XW^V$$

where $W^Q$, $W^K$, and $W^V$ represent the parameters learned by the model. For each attention head, the model computes attention scores and normalizes them using the softmax function

to derive the attention weights.

$$\text{head}_h = \text{Attention}\left(QW^{Qh}, KW^{Kh}, VW^{Vh}\right) = softmax\left(\frac{QW^{Qh}\left(KW^{Kh}\right)^T}{\sqrt{d_k}}\right)VW^{Vh}. \qquad (17)$$

Finally, the outputs from all attention heads are concatenated and then transformed *via* a linear layer to produce the final output of the MHSA layer.

$$MultiHead(Q, K, V) = Concat(head_1, head_2, \ldots, head_h)W^o. \qquad (18)$$

Where $W^o$ is a trainable weight matrix used for dimension adjustment.

At the output stage, a fully connected Dense layer integrated the features processed by the MHSA and employed a softmax activation function for three-class sentiment classification. This ensured that the output results were both probability distributions and mutually exclusive across categories. To mitigate overfitting and enhance the model's generalization ability, Dropout layers (*Chen et al., 2023a*) were incorporated between the BiLSTM layers and between the MHSA layer and the Dense layer. Dropout layers randomly deactivated a proportion of neuron connections during training, reducing complex co-adaptations and promoting the development of more robust features.

Regarding loss function selection, this study employed the sparse_categorical_crossentropy loss function, which was well-suited for multi-class classification problems with softmax activation. This function quantified the dissimilarity between the predicted probability distribution and the true probability distribution. The mathematical formula for categorical cross-entropy loss is:

$$loss = -\frac{1}{m}\sum_{j=1}^{m}\sum_{j=1}^{k} y_{ij}\log\left(\check{y}_{ij}\right). \qquad (19)$$

Where $m$ represented the number of samples, and $k$ represented the number of categories.

The Adam optimizer was selected for model optimization in this study. Adam utilizes adaptive learning rates and incorporates bias correction, resulting in a stable learning rate range across iterations. This enhanced the stability and convergence speed of model training.

The double-layer improved BiLSTM architecture ($L_t$) increased the model's capacity to represent and analyze complex text data effectively.

## Model training and validation

Dataset $E_t$ and BiLSTM model $L_t$ were utilized for model evaluation and optimization. The data were divided into training set $X_t$, and a test set $C_t$ to assess model performance and ensure generalizability.

To optimize the model's ability to process text and capture temporal dynamics, the model structure, parameters, and learning rate were iteratively refined. This process aimed to address the limitations of the LDA model and leverage the strengths of the fusion approach. Through analysis of performance metrics during training and referencing the training set performance report, optimal model parameters were identified and saved as a $Q_t$ file.

## Model evaluation and validation

The model was evaluated by loading the training dataset $X_t$ and the previously optimized model weights $Q_t$. Inferences were performed on the optimized model using the test dataset $C_t$, generating updated coherence and perplexity curves for visual assessment of model performance.

Key performance indicators (KPIs), including accuracy and F1 score (*Santhiran, Varathan & Chiam, 2024*), were calculated for both the original and improved models. Comparison of these KPIs demonstrates the enhanced performance of the improved model across multiple dimensions.

## EXPERIMENTS

The research was conducted utilizing a Windows 11 operating system environment. Hardware specifications included an Intel(R) Core (TM) i5-8265U CPU with a processing speed of 1.60 GHz (up to 1.80 GHz). The programming language platform employed was Python version 3.6.8. Software tools for development and analysis encompassed PyCharm Community Edition 2023.1.3 and Jupyter Notebook.

## Datasets

In this study, an in-depth analysis of online comment data from visitors was conducted. A total of 7,991 core comments were collected from Dianping.com (*Zhang & Long, 2024*) through the utilization of web crawling technology (*Zhang & Wu, 2022*). Additionally, publicly available datasets (comprising 8,186 Chinese product reviews (*IDataScience, 2022*) and 50,000 English movie and TV show reviews (*Ni, Li & McAuley, 2019*)) were used to validate the model's cross-lingual generalization capabilities. Given the correlation between star ratings and user sentiment, comments with 4 or 5 stars were classified as positive evaluations, those with 2 to 3.5 stars as neutral evaluations, and those with 1.5 stars or below as negative evaluations. The detailed classification is presented in Table 1.

In the data processing stage, regular expressions were first utilized to remove special symbols and extraneous punctuation from the text. Subsequently, the jieba library was employed for word segmentation, and stop words were eliminated. Following preprocessing, the dataset was randomly partitioned into a training set and a testing set at a ratio of 80%/20%. These sets were then used for model training and performance evaluation, respectively.

## Hyperparameter setting

The number of topics in the LDA model significantly influenced its performance. By plotting the coherence curve (Fig. 8A) and the perplexity curve (Fig. 8B) for each dataset, the optimal number of topics was determined. For the Shanghai Astronomy Museum dataset, coherence peaked at four topics, with perplexity (after inversion) also being relatively high, indicating the model's optimal generalization ability. For the mobile phone products dataset, coherence was highest at six topics, with good perplexity (after inversion). For the Amazon movie and TV review dataset, both coherence and perplexity (after inversion) reached their highest points at seven topics. The traditional LDA model

PeerJ Computer Science ______________________________

**Table 1   Data classification table.**

| Comment Source | Language | Sentence properties | Sentence number | Percentage |
|---|---|---|---|---|
| Movies and TV series on Amazon platforms | English | Positive | 42468 | 84.94% |
| | | Neutral | 5024 | 10.04% |
| | | Negative | 2508 | 5.02% |
| Mobile Phone Products | Chinese | Positive | 3042 | 37.16% |
| | | Neutral | 2487 | 30.38% |
| | | Negative | 2657 | 32.46% |
| Shanghai Astronomy Museum | Chinese | Positive | 7069 | 88.46% |
| | | Neutral | 600 | 7.51% |
| | | Negative | 322 | 4.03% |

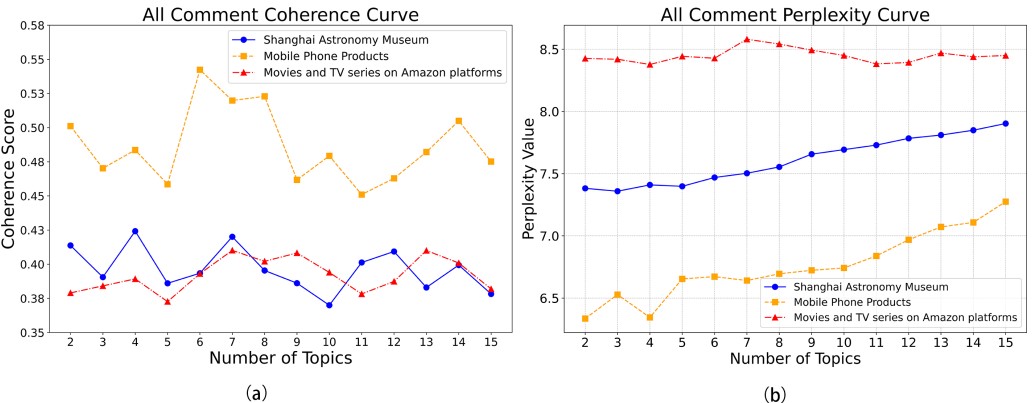

**Figure 8   (A) LDA consistency curves across datasets; (B) LDA perplexity curves across datasets.**

exhibited some overfitting when processing the Chinese datasets, with perplexity increasing as the number of topics rose. However, the exponential upward trend of the perplexity curve also hinted at the limitations of the LDA model in handling complex contexts and temporal sequences. Therefore, next studies may consider integrating other models to enhance overall performance and address the overfitting issue.

The number of topics in the LDA model was optimized by evaluating coherence and perplexity on the respective datasets. The RealNVP model utilized 4 flows to balance model complexity and expressiveness, and was trained using maximum likelihood estimation and the Adam optimizer. The double-layered BiLSTM model employed 128-dimensional word vectors, with 128 and 64 units in its respective layers, to achieve progressive feature abstraction. The model incorporated four attention heads to capture multifaceted semantic information, with the attention dimension aligned with the BiLSTM output dimension. During training, a dropout ratio of 0.5 and a batch size of 32 were employed. Categorical cross-entropy was used as the loss function, and the Adam optimizer was utilized for optimization. Other parameter settings were as follows: a random seed of 42, a document processing batch size of 100, and 15 corpus iterations.

## Evaluation metrics

For model evaluation, this study employed multiple metrics to comprehensively assess model performance. The effectiveness of model improvements was demonstrated by comparing coherence and perplexity curves.

To further quantify model performance, accuracy (ACC) (*Bello, Ng & Leung, 2023*) was selected as the primary evaluation metric. Accuracy measured the proportion of correctly classified instances out of the total number of instances. It was calculated as follows:

$$\text{Accuracy} = \frac{TP + TN}{TP + TN + FP + FN}. \tag{20}$$

Where *TP* and *TN* represented true positives and true negatives, respectively, indicating correctly predicted positive and negative instances. *FP* and *FN*, on the other hand, represented false positives and false negatives, which were incorrectly predicted positive and negative instances.

In addition to accuracy, the F1 score (*Hsieh & Zeng, 2022*) was included as a supplementary evaluation metric to account for both precision and recall. The F1 score was the harmonic mean of precision and recall, providing a balanced measure of a model's performance, particularly in cases of imbalanced class distributions (*Powers, 2011*). It was calculated as follows:

$$F1 = \frac{2 * TP}{2 * TP + FP + FN}. \tag{21}$$

By utilizing multiple evaluation metrics, including accuracy and F1 score, this study conducted a comprehensive analysis of model performance from various perspectives.

## RESULTS AND DISCUSSION

### Comparison of model performance results

This study compared and analyzed the performance of several models for sentiment analysis: the traditional LDA model, the LSTM model, the LDA-CNN model (*Liu et al., 2022*), the CNN-LSTM model (*Khan et al., 2022a*), and the proposed LDA-RealNVP-DBiLSTM-MHSA model. The LDA-CNN model enhanced short text classification by integrating global topic information with local contextual features. The CNN-LSTM model has demonstrated strong performance in sentiment analysis of English and Roman Urdu social media texts. This study addressed the challenges of text relevance and temporality in sentiment analysis by optimizing topic selection and model structure, incorporating RealNVP with Shuffle operation to enhance LDA topic features, utilizing a double-layered improved BiLSTM to model text temporality, and integrating MHSA to capture multi-scale semantic relationships.

To objectively evaluate model performance, all models were trained and tested on the same dataset (80% training set, 20% testing set). Comparing coherence and perplexity across models (Fig. 9) revealed consistent improvements in coherence for the optimized LDA-RealNVP-DBiLSTM-MHSA model across all datasets, with a notable improvement of approximately 10% on the Shanghai Astronomy Museum dataset. Additionally, the model demonstrated good performance in terms of perplexity. The incorporation of strategies
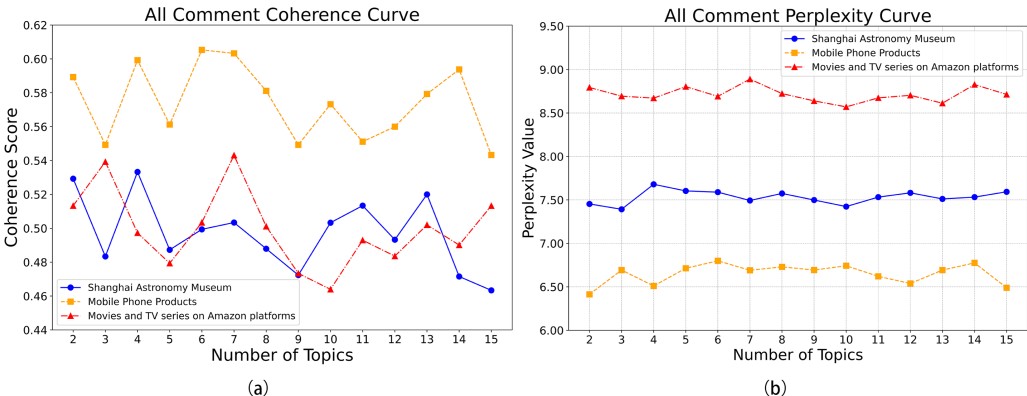

**Figure 9** (A) LDA-RealNVP-DBiLSTM-MHSA consistency curves across datasets; (B) LDA-RealNVP-DBiLSTM-MHSA perplexity curves across datasets.

**Table 2** Comparison of 80% training set model evaluation metrics.

| Comment Source | Model | ACC | F1 |
|---|---|---|---|
| Movies and TV series on Amazon platforms | LSTM | 86.41% | 85.31% |
| | LDA-CNN | 85.12% | 87.92% |
| | CNN-LSTM | 86.52% | 87.78% |
| | LDA-RealNVP-DBiLSTM-MHSA | 89.33% | 88.81% |
| Mobile Phone Products | LSTM | 86.33% | 85.21% |
| | LDA-CNN | 82.56% | 81.09% |
| | CNN-LSTM | 78.31% | 77.93% |
| | LDA-RealNVP-DBiLSTM-MHSA | 87.03% | 85.91% |
| Shanghai Astronomy Museum | LSTM | 87.14% | 85.19% |
| | LDA-CNN | 87.48% | 83.94% |
| | CNN-LSTM | 81.13% | 79.92% |
| | LDA-RealNVP-DBiLSTM-MHSA | 88.62% | 87.86% |

such as RealNVP with Shuffle operation, the double-layer improved BiLSTM, and MHSA effectively enhanced the overall performance of the model in handling text. This mitigated the limitations of the traditional LDA model in modeling complex contextual temporality and also alleviated the issue of overfitting.

The results of the multi-model performance comparison were presented in Table 2. The optimized LDA-RealNVP-DBiLSTM-MHSA model outperformed the LSTM, LDA-CNN, and CNN-LSTM models in terms of both ACC and F1 score. Notably, while the CNN-LSTM model has demonstrated strong performance in processing English and Roman Urdu data, its performance was comparatively lower on the Chinese dataset used in this study, as reflected in its ACC and F1 scores. LSTM outperforms LDA-CNN in online comment sentiment analysis primarily because LSTM can effectively capture long-range

**Table 3  Comparison of 20% testing set model evaluation metrics.**

| Comment source | Model | ACC | F1 |
|---|---|---|---|
| Movies and TV series on Amazon platforms | LSTM | 85.21% | 83.96% |
| | LDA-CNN | 84.65% | 87.07% |
| | CNN-LSTM | 85.41% | 86.78% |
| | LDA-RealNVP-DBiLSTM-MHSA | 88.76% | 88.09% |
| Mobile phone products | LSTM | 85.07% | 83.86% |
| | LDA-CNN | 82.19% | 80.72% |
| | CNN-LSTM | 76.95% | 75.89% |
| | LDA-RealNVP-DBiLSTM-MHSA | 86.12% | 84.16% |
| Shanghai Astronomy Museum | LSTM | 87.17% | 83.10% |
| | LDA-CNN | 87.74% | 84.13% |
| | CNN-LSTM | 80.12% | 79.56% |
| | LDA-RealNVP-DBiLSTM-MHSA | 87.93% | 87.37% |

dependencies and contextual information in sequential data, thereby better understanding complex sentence structures and emotional expressions.

## Comparison of experimental results

To process the comment data more effectively, the study employed Skip-gram and the LDA model to reduce the dimensionality of high-dimensional word vectors. RealNVP was then utilized for feature enhancement, and the resulting vectors were input into a double-layer improved BiLSTM and MHSA model for refined processing. The experimental results, as shown in Table 3, demonstrate the exceptional performance of the proposed LDA-RealNVP-DBiLSTM-MHSA model in processing online review data. It significantly outperformed other models in both accuracy and F1 evaluation metrics. This outcome confirms the effectiveness and practicality of the LDA-RealNVP-DBiLSTM-MHSA model, particularly in handling complex text relevance and temporality.

To further illustrate the advantages of the LDA-RealNVP-DBiLSTM-MHSA model, Fig. 10 presents comparisons of ACC and F1 values for various models across different datasets. The results consistently confirmed the superiority of the proposed model over the baseline models in terms of all evaluation metrics. On the English review dataset, as shown in Fig. 10B, although the F1 value of the proposed model was close to that of the LDA-CNN model, it still achieved the highest score. This can be attributed to the incorporation of RealNVP with Shuffle operation after LDA in the proposed model, which enabled it to better capture text relevance. The experimental results fully substantiate the superiority of the LDA-RealNVP-DBiLSTM-MHSA model in processing both Chinese and English online review data.

## Discussion

The LDA-RealNVP-DBiLSTM-MHSA model proposed in this study outperformed existing LDA fusion models and LSTM fusion models in both accuracy and F1 score.

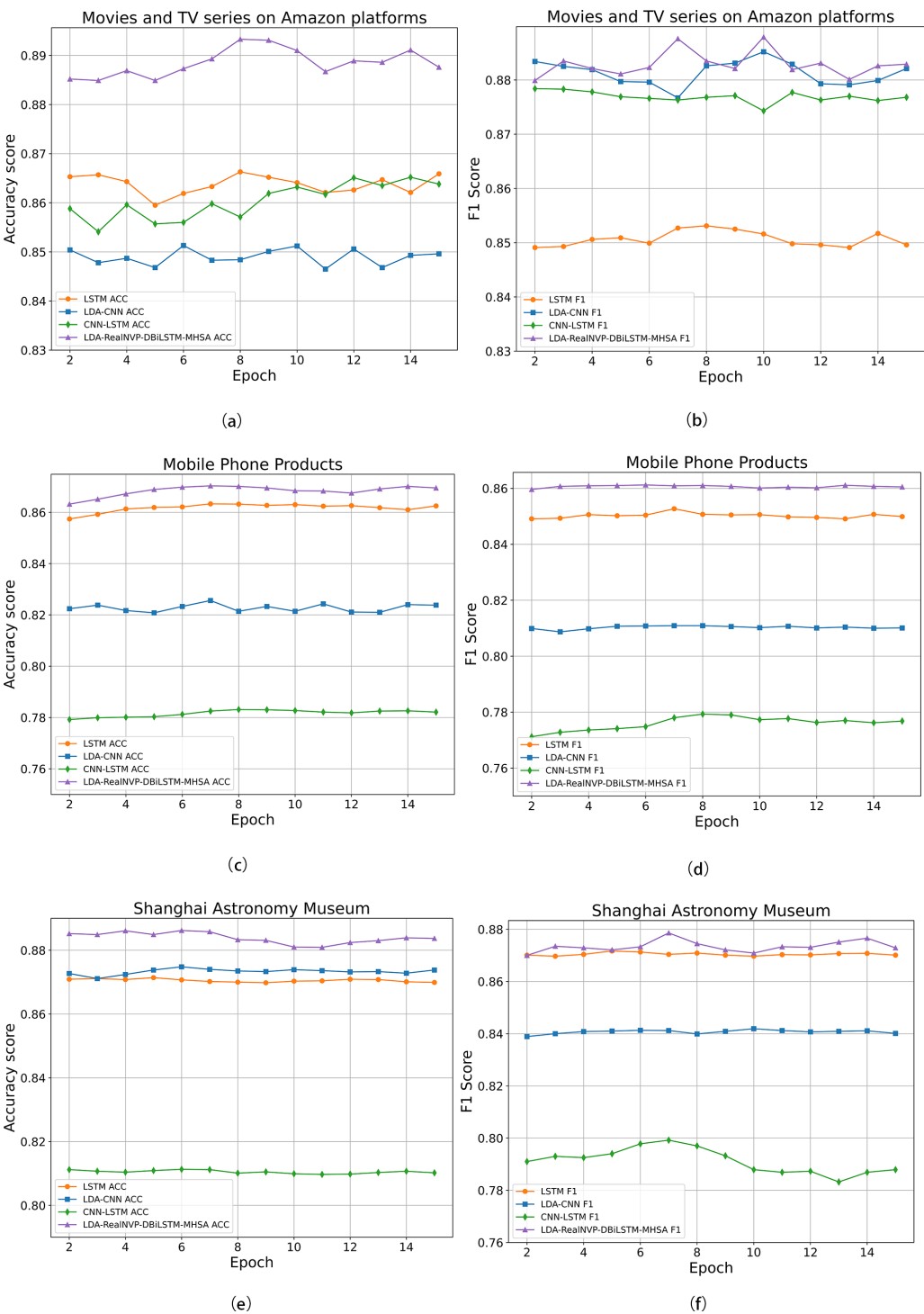

**Figure 10** (A) Accuracy on Amazon movie and TV show comments; (B) F1 score on Amazon movie and TV show comments; (C) Accuracy on mobile phone product comments; (D) F1 score on mobile phone product comments; (E) Accuracy on Shanghai Astronomy Museum comments.

At the level of LDA fusion, building upon previous research (*He, Zhou & Zhao, 2022*; *Liu et al., 2022*; *Wang, Sun & Wang, 2022*; *Watanabe & Baturo, 2024*; *Wu & Shen, 2024*; *Wu, Du & Lin, 2023*), this study further enhanced the analysis of text relevance and temporality. Unlike previous approaches that directly input LDA topic features into subsequent models, this study innovatively introduces a Shuffle-enhanced RealNVP to deeply express and enhance LDA topic features. RealNVP maps high-dimensional, sparse topic features to a continuous, low-dimensional latent space through reversible transformations. The Shuffle operation further breaks local correlations, enhancing the model's ability to model long-range dependencies in high-dimensional complex data, thereby improving the robustness of feature representation. This direct concatenation approach not only compensates for the shortcomings of traditional LDA models in capturing key semantic information such as word order and subtle differences in word meaning, but also ensures the preservation of fine-grained semantic and high-level topic information of the text, thereby effectively enhancing the relevance of the text. Compared to the LDA-CNN model (*Liu et al., 2022*), this model achieved a significant improvement in accuracy. This improvement was primarily attributed to the deep optimization of LDA topic features by the Shuffle-enhanced RealNVP, as well as the effective temporal modeling and semantic association capture of the enhanced features by the double-layer BiLSTM and MHSA.

In the realm of LSTM fusion models, building upon prior research on time series analysis (*Liu, Wang & Li, 2023*), topic embedding (*Xuan & Deng, 2023*), CNN and LSTM fusion (*Khan et al., 2022a*), and the combination of LDA and single-layer LSTM (*Zeng, Yang & Zhou, 2022*), this study adopted a Shuffle-enhanced RealNVP, a double-layer BiLSTM structure integrating forget gates and input gates, and MHSA. The progressive processing mechanism of the double-layer BiLSTM enables a deeper exploration of the temporal features and long-term dependencies embedded within the RealNVP-enhanced feature vectors, effectively addressing the issue of text temporality and mitigating overfitting. MHSA, by simultaneously attending to different aspects and granularities of the input sequence, achieves a balance between global semantics and local details, resulting in significant advancements in semantic understanding.

The proposed LDA-RealNVP-DBiLSTM-MHSA model, with its core innovation being the introduction of Shuffle-enhanced RealNVP, effectively improved the accuracy and F1 score of sentiment analysis on online review texts. By integrating LDA's topic modeling capabilities, the enhanced RealNVP, and the enhanced sequence processing abilities of the double-layer BiLSTM and MHSA, the model effectively addressed the overfitting issue of traditional LDA models on Chinese comment data, enhanced the capture of text relevance and temporality, and provided a novel approach and methodology for Chinese text sentiment analysis.

## CONCLUSIONS

Addressing the limitations of traditional sentiment analysis models in handling text relevance and temporality, this study proposed a method integrating LDA, Shuffle-enhanced RealNVP, a double-layer improved BiLSTM, and MHSA, effectively improving

the efficacy of text sentiment analysis. By introducing the Shuffle mechanism to enhance RealNVP, a deep expression and enhancement of LDA topic features was achieved, providing higher-quality input for subsequent temporal modeling. Simultaneously, by coupling the LSTM's forget gate and input gate, the processing rate was appropriately mitigated while maintaining accuracy. This approach proved particularly suitable for datasets with a predominance of positive comments, fully leveraging BiLSTM's advantages in temporal data processing. Compared to traditional models such as LDA, LSTM, LDA-CNN, and CNN-LSTM, it demonstrated superior performance in terms of accuracy, F1 score, and other metrics.

The proposed LDA-RealNVP-DBiLSTM-MHSA fusion model exhibited advantages in accurately capturing comment topic information. It could extract valuable sentiment information from massive user comments more quickly and accurately, providing precise decision support for brand reputation management, market trend forecasting, public policy formulation, and venue design improvement, ultimately contributing to the continuous improvement of user satisfaction and the sustainable development of data resources.

However, the processing of high-dimensional Chinese comment data using this method still presented certain challenges. Significant memory consumption and longer processing time somewhat limited its application to ultra-large Chinese comment datasets. Additionally, variants of the LSTM structure might not be universally suitable. While the coupling design performed exceptionally well on specific tasks, it might not be applicable to others. For instance, in scenarios requiring precise control over information retention and addition, traditional independent gate designs might prove advantageous.

Therefore, this study proposes the following directions for future work:

(1) Enhancing the Skip-gram model: Incorporating deeper linguistic features, such as part-of-speech tags and dependency relations, into the Skip-gram model to enrich word vector representations and more comprehensively capture the semantic nuances of text.

(2) Optimizing network structure: Investigating the replacement of fully connected layers in the LSTM with locally connected layers to reduce the number of parameters and computational complexity, potentially mitigating memory constraints and improving processing speed.

(3) Introducing multimodal features: Considering the fusion of multimodal features such as text, audio, and video, and adopting new fusion methods to further improve the accuracy and efficiency of sentiment analysis.

### Funding

This research was funded by Industry-university cooperation and collaborative education project of the Ministry of Education (Grant Number: 231003221260825). The funders had no role in study design, data collection and analysis, decision to publish, or preparation of the manuscript.

## Grant Disclosures

The following grant information was disclosed by the authors:

Industry-university cooperation and collaborative education project of the Ministry of Education: 231003221260825.

## Competing Interests

Gang Chen is an employee of the Infrastructure Technology Business, Ant Group.

## Author Contributions

- Yan Zhu conceived and designed the experiments, performed the experiments, analyzed the data, prepared figures and/or tables, authored or reviewed drafts of the article, and approved the final draft.
- Rui Zhou performed the experiments, analyzed the data, performed the computation work, prepared figures and/or tables, authored or reviewed drafts of the article, and approved the final draft.
- Gang Chen conceived and designed the experiments, authored or reviewed drafts of the article, and approved the final draft.
- Baili Zhang analyzed the data, authored or reviewed drafts of the article, and approved the final draft.

## Data Availability

The data and code are available in the Supplemental Files.

The dataset used in the study is available at: IDataScience. 2022. Chinese Product Reviews. Available at http://www.idatascience.cn/dataset-detail?table_id=100936 (accessed 09.10 2024).

## Supplemental Information

Supplemental information for this article can be found online at http://dx.doi.org/10.7717/peerj-cs.2542#supplemental-information.

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
