# Peer review of "Enhancing sentiment analysis of online comments: a novel approach integrating topic modeling and deep learning"

_PeerJ Computer Science, doi:10.7717/peerj-cs.2542_

## Round 0.1 · original submission · Major Revisions

The current version does not reach the criteria of the journal. I hope the authors carefully consider the comments and revise the work accordingly. We look forward to your revision

· Appeal

Appeal


· · Academic Editor

Reject

Thanks to the authors for their submission. However, the reviewer pointed out that the contribution of this work is limited and the dataset is small. I cannot be more positive this time. I hope this decision will not frustrate you.

Reviewer 1 ·

Basic reporting

The research presented in this manuscript proposed the integration of LDA with the LSTM network for analysing the sentiments or opinions of comments from visitors to the Shanghai Astronomy Museum. The integration of both methods for text classification tasks is not new, which can be found in several pieces of literature. Thus, the proposed approach's novelty can be appreciated but is relatively minimal. The specific problem that the research aims to address is rather vague and not convincingly discussed in the manuscript. The authors mentioned in the manuscript the challenges of text relevance and temporality in sentiment analysis. However, how temporality is tackled and subsequently evaluated in the research diminished. The treatment and subsequent evaluation of temporality in this study are inadequate, failing to fully address the temporal aspects of the research question. This limitation significantly weakens the overall findings and conclusions drawn from the data. When dealing with the issues of temporality, it is essential to highlight how the proposed method in this manuscript deals with dynamic text-based sentiment classification and reduces misclassification costs.
Most of the methods presented in the manuscript are well-known and basic techniques that have been discussed in many literature and textbooks.
Nevertheless, the manuscript is well-written and adheres to the Peerj standards.

Experimental design

The authors evaluated their proposed method against three established models: the basic LSTM, LDA-CNN, and CNN-LSTM. The experimental dataset comprised 7,991 comments sourced from Dianping.com. While the proposed method demonstrated superior performance in terms of accuracy and F1 metrics compared to the baseline models, the study's generalizability is limited. The dataset's relatively small size and narrow focus on a single domain restrict the broader applicability of the findings. The authors should consider larger, more diverse datasets to validate the method's efficacy across multiple domains.

Validity of the findings

The findings are limited to the Shanghai Astronomy Museum online review texts. Therefore, the findings cannot be generalised to other domains. The authors need to consider different and larger datasets.

Additional comments

In general, the manuscript has a high level of writing proficiency and is readily comprehensible. While the suggested method is commendable, it lacks originality and comprehensive experimentation. The aspect of temporality, as frequently highlighted in the manuscript, was not addressed convincingly.

·

Basic reporting

The paper is written in clear English. Most of the sections are written in a language that is understandable. However some sentences are too long. For instance line 17 is too long. I propose that it should be split into two sentences. Mentioning of the dataset used in line 18 I find it unnecessary since the model developed can be used with other comments in Chinese. Unless the authors make it clear that the model can be used for Shanghai Astronomy Museum comments only. The mention of dataset is properly done in line 24 and 25.
The introduction is well written giving the context of the research. However, Shanghai Astronomy Museum is being mentioned too early in the introduction (line 32). I feel that this should come much later after describing the generic issues/problems. Actually, I propose that the introduction starts with the sentence in line 35 to line 38.
The problem is well articulated in the introduction. The shortcomings of the traditional sentiment analysis approaches are well explained and the motivation behind the development of the proposed model. However, the author did not explain clearly why Latent Dirichlet Allocation (LDA) is limited in handling temporal dynamics in sentiment expressions. The author needs to provide an explanation after Line 58 and 59 showing why LDA is limited.
The author has explained why the skip-gram model was preferred in line 227, 228 and 229. To support the argument there is need to provide appropriate citation. The methodology which is the proposed approach is well presented. However, the author should state clearly how the word vector (Dt) generated from the skip-gram and the document-matrix (Ct) generated from TF-IDF are used as input into the LDA. There should be coherence between the explanation in the methodology and figure 4.

Experimental design

The experimental design presented is sufficient. The author has describe the dataset used, the preprocessing approaches employed, the parameters used in the deep learning model and the model evaluation techniques and metrics which include the accuracy and F1 measure. However, the author should have supported the choice of the evaluation metrics with appropriate citation especially in line 437 to 448.

Validity of the findings

The results are presented by the author in tables and charts. However, I feel the author can enhance the presentation of the results and discussion of the same. For instance, the author has introduced figure 9 in line 487. Consequently, the author has not discussed the results presented in the figure. I propose that the author adopts a format in which he introduces the results, presents them in a table or chart, then discusses the results presented. An explanation on why CNN is not performing well as compared to LSTM when combined with LDA can improve the discussion of the results.

Additional comments

The article is fairly done. I propose that it can be a good article for publication if the revisions are done before publication.

---

## Round 0.2 · accepted · Accept

Thanks to the authors for their efforts to improve the work. The authors considered the comments carefully and successfully addressed the reviewers' concerns. The only issue proposed by the reviewer can be revised in the proof stage. It can be accepted now. Congrats!

Reviewer 1 ·

Basic reporting

The authors have thoroughly reviewed and addressed all comments and concerns raised in the previous version of this manuscript. Each point has been carefully considered, and revisions have been made to enhance the clarity, accuracy, and overall quality of the work in response to the feedback received.

Experimental design

The experiment has been revised to include new datasets (Amazon movies and news, Chinese mobile phone reviews), enhancing the robustness and validity of the findings. By incorporating a broader range of data, the study now provides stronger justification for the generalizability of its results across diverse contexts and conditions. This expanded approach allows for a more comprehensive evaluation of the findings' applicability beyond the original dataset, offering greater confidence in the conclusions drawn.

Please cite accordingly the two new used datasets.

Validity of the findings

Overall good based on the two new datasets.

Additional comments

The authors must cite the two new datasets used in the experiment.

·

Basic reporting

The paper is more improved. The authors have been keen in revising the manuscript as per the comments of the reviewer.

Experimental design

The experiments presented are clear and appropriate for the research.

Validity of the findings

The findings are well presented and clear.

Additional comments

No additional comment